# Semaglutide Enhances Cellular Regeneration in Skin and Retinal Cells In Vitro

**DOI:** 10.3390/pharmaceutics17091115

**Published:** 2025-08-27

**Authors:** Ioanna A. Anastasiou, Anastasios Tentolouris, Panagiotis Sarantis, Athanasia Katsaouni, Eleni Rebelos, Iordanis Mourouzis, Constantinos Pantos, Nikolaos Tentolouris

**Affiliations:** 1Department of Propaedeutic Internal Medicine, ‘Laiko’ General Hospital, Medical School, National and Kapodistrian University of Athens, 11527 Athens, Greece; antentolouris@hotmail.com (A.T.); eleni.rebelos@utu.fi (E.R.); 2Department of Pharmacology, Medical School, National and Kapodistrian University of Athens, 11527 Athens, Greece; athanasia.paraskevikatsaouni@gmail.com (A.K.); imour@med.uoa.gr (I.M.); cpantos@med.uoa.gr (C.P.); 3Molecular Oncology Unit, Department of Biological Chemistry, Medical School, National and Kapodistrian University of Athens, 11527 Athens, Greece; panayotissarantis@gmail.com

**Keywords:** semaglutide, oxidative stress, wound healing, cellular regeneration, diabetic complications

## Abstract

**Background/Objectives:** Glucagon-like peptide-1 (GLP-1) is an endogenous hormone with receptors widely expressed across multiple organs. GLP-1 receptor agonists (GLP-1RAs), primarily used for diabetes management, have demonstrated anti-inflammatory and antioxidant properties beyond glucose regulation. This study explores the protective effect of semaglutide, a GLP-1RA, in reducing oxidative stress and promoting wound healing in human dermal fibroblasts. Additionally, it assesses whether semaglutide offers the direct protection of retinal endothelial cells under oxidative stress. **Methods:** Human dermal fibroblasts and retinal endothelial cells were treated with semaglutide at concentrations ranging from 0 to 45 pg/mL for 24 h under oxidative stress induced by hydrogen peroxide (H_2_O_2_). Cell viability and ATP levels were measured via MTT and ATP assays. Apoptosis was evaluated using propidium iodide staining. Intracellular reactive oxygen species (ROS) and mitochondrial superoxide were assessed through confocal microscopy with specific fluorescent probes. Wound healing was tested using scratch assays, with closure monitored over time and quantified with ImageJ (version 1.51). Gene expression levels of antioxidants, extracellular matrix components, inflammatory cytokines, and MMPs (MMP3, MMP9) were determined via real-time PCR. **Results:** Semaglutide significantly improved cell viability and ATP production under oxidative stress (*p* < 0.001), while reducing apoptosis and intracellular ROS levels. It notably accelerated fibroblast wound closure, achieving near-complete restoration. Gene analysis revealed increased expression of antioxidant and ECM-related genes, along with decreased pro-inflammatory cytokines and MMPs, indicating reduced inflammation and enhanced tissue remodeling. **Conclusions:** Semaglutide offers robust antioxidative and cytoprotective effects in dermal fibroblasts and retinal endothelial cells, promoting wound healing. These findings highlight its therapeutic potential for diabetic foot ulcers and diabetic retinopathy, supporting further in vivo investigation.

## 1. Introduction

Glucagon-like peptide-1 (GLP-1) is an incretin hormone predominantly involved in glucose regulation, exerting its effects by stimulating insulin secretion and suppressing glucagon release [1,2]. The therapeutic potential of GLP-1 has driven the development of GLP-1 receptor agonists (GLP-1RAs), which emulate its physiological effects [3]. These pharmacological agents have been shown to be effective in the management of chronic diseases such as type 2 diabetes (T2D) and obesity, by improving glycemic control, inducing weight loss, and decreasing cardiovascular risk factors [4]. Clinical trial data consistently demonstrate their efficacy, including significant reductions in adiposity and enhancements in the metabolic rate among patients with obesity [5].

T2D is often complicated by chronic renal disease, neuropathy, diabetic retinopathy (DR), and cardiovascular disease [6,7]. Oxidative stress plays a key role in the development of these complications by damaging molecules and triggering pathways such as apoptosis, inflammation, and vascular dysfunction [7,8,9,10,11,12,13]. Excess free radicals increase inflammatory mediators (e.g., transforming growth factor β (TGF-β) and tumor necrosis factor α (TNF-α) and promote cell death through mechanisms involving p53 and caspases, leading to tissue damage [14]. Oxidative stress directly damages lipids, proteins, and nucleic acids, impairing cellular functions and lipid homeostasis, thereby exacerbating cardiovascular complications [15]. Managing oxidative stress could therefore be vital in preventing or reducing diabetic complications [16].

Recent studies have illuminated additional roles of GLP-1 in various physiological processes, notably its potential influence on oxidative stress and tissue repair mechanisms [1,17,18]. Oxidative stress, an imbalance between reactive oxygen species (ROS) production and antioxidant defenses, is a critical factor that hampers effective wound healing [19]. Elevated levels of oxidative stress can cause cellular damage, promote inflammation, and delay tissue regeneration, particularly in individuals with metabolic disorders such as T2D [20].

Moreover, increased oxidative stress accelerates the formation of advanced glycation end-products (AGEs), which are produced through the non-enzymatic glycation of proteins, lipids, and nucleic acids [21,22]. The accumulation of AGEs in tissues exacerbates oxidative damage and disrupts normal tissue repair processes. This cascade not only impairs wound healing but also contributes to the progression of diabetic complications. Together, the interplay among oxidative stress, AGEs, and impaired tissue regeneration creates a detrimental cycle that hinders recovery and perpetuates tissue dysfunction in metabolic disease states [21,22].

Emerging research suggests that GLP-1 may exert antioxidative effects by reducing ROS levels, enhancing antioxidant enzyme activity, and modulating inflammatory responses [1,23,24]. Concurrently, studies have indicated that GLP-1 can influence critical aspects of wound healing, such as promoting angiogenesis, fibroblast proliferation, and extracellular matrix formation. Understanding the molecular pathways through which GLP-1 impacts oxidative stress and wound repair can open new avenues for therapeutic intervention, particularly in populations with impaired healing capacity [1,23,24].

In the SUSTAIN 6 trial, a two-year, pre-approval cardiovascular outcomes study, semaglutide was associated with a statistically significant increase in the risk of DR complications relative to the placebo [25]. Nonetheless, this association was not corroborated by findings from other concurrent studies [26].

The primary objective of this study was to examine whether semaglutide has cytoprotective effects against oxidative stress in human dermal fibroblasts (NHDFs) and human retinal endothelial cells (HRECs). Specifically, the study aimed to assess the effect of semaglutide on key aspects of wound healing in fibroblasts, including cell migration and the expression profiles of extracellular matrix-associated genes. Additionally, the research sought to determine whether semaglutide enhances cellular viability, attenuates apoptosis, and mitigates oxidative damage in response to hydrogen peroxide-induced oxidative stress. Furthermore, the study examined whether semaglutide exerts direct protective effects on retinal endothelial cells, thereby suggesting a broader potential therapeutic role in DR that extends beyond its established glycemic regulation. These findings aim to contribute to a deeper understanding of semaglutide’s cellular mechanisms and its prospective application in vascular and tissue repair within diabetic retinal pathology.

## 2. Material and Methods

### 2.1. Cell Culture

Human dermal fibroblasts (NHDFs) isolated from normal adult human skin were purchased from Lonza Clonetics™ (Catalog No. CC-2511, Lonza Walkersville, MD, USA). NHDF cells were cultured in accordance with the manufacturer’s instructions. They were maintained in Fibroblast Growth Basal Medium (FGM™; Catalog No. CC-3131, Lonza Walkersville, USA). Cells were passaged once they reached 70–80% confluence, using the ReagentPack™ (Catalog No. CC-5034, Lonza Walkersville, USA) for subculturing.

Human retinal endothelial cells (HRECs; Catalog No. P10880, Innoprot, Bizkaia, Spain) were cultured under standard conditions to ensure their viability and functionality. Cells were seeded in tissue-culture-treated flasks and maintained in endothelial cell growth medium (EGM-2), supplemented with endothelial growth factors, including vascular endothelial growth factor (VEGF), hydrocortisone, basic fibroblast growth factor (bFGF), and insulin-like growth factor-1 (IGF-1), as provided by the manufacturer, to promote endothelial proliferation and preserve the phenotype.

Briefly, both NHDF and HREC cells were pre-incubated with hydrogen peroxide (H_2_O_2_) at a concentration of 100 μM for 12 h to induce oxidative stress [27]. Following treatment, cells were washed thoroughly with 1× Phosphate-Buffered Saline (PBS; Catalog No. L0615-1000, Biowest, Business Park Ln, Riverside, MO, USA). Then, NHDF and HREC cells were incubated in the presence of different concentrations of semaglutide (11.25, 22.5 and 45 pg/mL) for 24 h. The control consisted of fresh culture medium administered after exposure to oxidative stress conditions (100 μM H_2_O_2_). For experiments, cells were seeded at the density of 6.0 × 10^3^ cells/well in 96-well plates, 2.5 × 10^5^ cells/well in 24-well plates and in eight-chamber slides, and 1.0 × 10^6^ cells/well in 6-well plates. The semaglutide concentrations used in this study were based on those reported in a previous study [28,29]. Semaglutide (≥95% purity, code 29969, Cyaman Chemical, Ann Arbor, MI, USA) was dissolved in 1 mg/mL DMSO (catalog 25300-062, Thermo Scientific, Fremont, CA, USA) due to its limited water solubility. The final concentration of DMSO in the culture medium was maintained at 0.1% *v*/*v*. The H_2_O_2_ stock solution was prepared at 30% (catalog 822287, Sigma-Aldrich, St. Louis, MO, USA) and diluted in the culture medium to a final concentration of 100 μM. All cell culture materials were sourced from Falcon (VWR International Eurolab, Llinars del Vallès, Spain). All other chemicals used in the study were of the highest available purity and obtained from reputable commercial suppliers.

### 2.2. MTT Measurement

NHDF and HREC cells were seeded in 96-well plates with six biological replicates, following the previously described procedure. At the end of the treatment period, MTT reagent (#30006, Biotium Inc., Fremont, CA, USA) was added to each well, and the assay was conducted according to the established protocol [30].

### 2.3. Cell Counting

Six biological replicates of NHDFs and HRECs were seeded in 96-well plates, following the previously described protocol. Cells were imaged at 24 h using phase contrast microscopy at 5× magnification. To assess cell viability, the Cell Counting Kit (30 dual-chambered slides, 60 counts) with trypan blue (#1450003, Bio-Rad Laboratories, Hercules, CA, USA) was used to distinguish live from dead cells, following the manufacturer’s instructions. Dead cells were quantified using a TC20 Automated Cell Counter (Bio-Rad Laboratories).

### 2.4. ATP Measurement

NHDF and HREC cells were seeded in 96-well plates with six biological replicates, following the previously described protocol. At the end of the treatment, cellular ATP levels were measured using the ATP Vialight Plus Assay Kit (LT07-12, Lonza, Switzerland), with ATP standards (LT27-008, Lonza) utilized for calibration, according to the standard procedures [30].

### 2.5. Detection of Intracellular Reactive Oxygen Species Level

Intracellular ROS levels were assessed using the DCFDA assay kit (ab113851, Abcam, Cambridge, MA, USA) following the manufacturer’s instructions. NHDF and HREC cells were cultured on 8-chamber slides as previously described. Cells were incubated with 25 μM DCFDA, diluted in phenol red-free Dulbecco’s Modified Eagle Medium supplemented with 15% FBS, for 45 min at 37 °C in a humidified incubator. After incubation, nuclei were stained with DAPI (1:1000 dilution in PBS, D9542, Sigma-Aldrich). ROS-induced green fluorescence was visualized with a confocal microscope (Olympus FluoView FV1000, Olympus, Tokyo, Japan), using excitation at 488 nm and emission at 525 nm. Fluorescence intensity was quantified using ImageJ software (https://imagej.nih.gov/ij/, accessed on 10 June 2025), with at least three images captured per treatment group to ensure accurate analysis.

### 2.6. Detection of Mitochondrial Superoxide

Mitochondrial superoxide production was evaluated using the MitoSOX™ Red mitochondrial superoxide indicator (M36008, Molecular Probes, UK) for live-cell imaging. NHDF and HREC cells cultured on 8-chamber slides were treated with semaglutide for 24 h, with three biological replicates conducted to ensure reliability. Post-treatment, MitoSOX Red fluorescence was imaged with a confocal microscope (Olympus FluoView FV1000, Olympus, Tokyo, Japan), using excitation at 510 nm and emission at 580 nm. Nuclei were counterstained with DAPI (1:1000 in PBS, D9542, Sigma-Aldrich), and images were captured at least three times per condition. Fluorescence intensity was quantified using ImageJ software (https://imagej.nih.gov/ij/, accessed on 10 June 2025), providing an accurate representation of mitochondrial superoxide levels.

### 2.7. AGE and RAGE Measurements

The levels of AGEs and RAGE in the cell culture supernatant were quantified using commercial ELISA kits (human AGE, EH0622; human RAGE, EH0408; Fine Test, Wuhan, China), following the manufacturer’s instructions. Cells seeded at 8000 per well in 24-well plates (with three replicates per treatment) were treated with semaglutide for 24 h and then harvested and homogenized in 500 μL of assay buffer on ice for 1 h. The homogenates were centrifuged at 10,000 rpm for 10 min to pellet insoluble debris. The soluble fraction was analyzed directly via ELISA. Standard curves were prepared within the specified range provided by the manufacturer, demonstrating good linearity. Absorbance was measured at 450 nm using a Multiskan™ FC Microplate Photometer (Thermo Scientific, USA), and concentrations of AGEs and RAGE were calculated based on these standard curves.

### 2.8. Propidium Iodide Staining Used for Detecting Apoptosis

NHDF and HREC cells were seeded in 6-well plates in triplicate and then treated with semaglutide for 24 h. Following treatment, cells were harvested and resuspended in 500 μL of PBS. To induce cell permeabilization, 4.5 mL of 80% ethanol was added to the cell suspension, and samples were incubated at 4 °C for 20 min. Cells were then centrifuged at 200× *g* for five minutes at 4 °C, and the supernatant was carefully removed. The cell pellet was washed twice with cold PBS, with each wash followed by centrifugation at 200× *g* for five minutes to remove residual ethanol.

Next, cells were permeabilized via incubation with 0.1% Triton X-100 for five minutes, followed by centrifugation to remove the supernatant. The cells were resuspended in 100 μL of propidium iodide (PI) staining solution, consisting of PI (4 μg/mL), ribonuclease (0.1 mg/mL), Tris-HCl buffer (pH 7.5, 100 mM), NaCl (150 mM), CaCl_2_ (1 mM), MgCl_2_ (0.5 mM), and NP-40 (0.1%). The stained cells were incubated at room temperature for 30 min in the dark to prevent photobleaching.

Following incubation, 400 μL of PBS was added to each sample, and flow cytometric analysis was performed using a BD FACSCalibur (BD Biosciences, San Jose, CA, USA). Data were acquired from the FL2 channel, and the percentage of PI-positive (i.e., apoptotic or dead) cells was calculated as a proportion of the total cell population: (PI-positive cells/total cells) × 100%.

### 2.9. In Vitro Wound Scratch Assay

A wound healing assay was conducted to assess the effect of semaglutide on NHDF cells using the scratch method. NHDF cells were seeded into each well of a 24-well plate and incubated at 37 °C in a humidified atmosphere containing 5% CO_2_ until a confluent monolayer was formed. A linear scratch was created with a sterile P200 pipette tip, and cell debris was removed by washing with PBS 1×. The initial wound was imaged at 0 h using phase contrast microscopy at a 5× magnification, prior to treatment with semaglutide. Subsequent images were captured at 3, 6, 12, 24, and 36 h post-treatment to monitor wound closure. The extent of healing was quantified as a percentage of the original wound area using ImageJ software, with wound closure calculated by comparing each time point to the initial 0 h image [30]. All experiments were conducted with three independent biological replicates.

### 2.10. RNA Preparation and Quantitative Real-Time PCR

Total RNA (500 ng) was extracted from a 6-well plate using the NucleoSpin RNA Kit (Macherey–Nagel, Düren, Germany), following the manufacturer’s instructions. The integrity of the eluted RNA (1 μL) was verified by running it on a 1% agarose gel, and the RNA concentration was measured spectrophotometrically at 260 nm using a NanoDrop spectrophotometer. Complementary DNA (cDNA) was synthesized using the Prime-Script RT Reagent Kit (Takara Bio, Shiga, Japan). Quantitative real-time PCR (RT-qPCR) was performed on a StepOnePlus™ Real-Time PCR System (Applied Biosystems™, Waltham, MA, USA) according to the manufacturer’s protocol. Primer sequences and gene characteristics are detailed in Appendix A, with primers designed as previously described [30]. Relative gene expression levels were calculated using the 2^−ΔΔCt^ method [30] with actin beta (ACTB) serving as the reference gene, as its expression is unaffected by the experimental treatments [31]. Data analysis was conducted with three biological replicates.

### 2.11. Graph Presentation and Statistical Analysis

All data are expressed as the mean ± standard deviation (SD). Graphs were generated using SigmaPlot 12.2 (Systat Software Inc., San Jose, CA, USA). Statistical analyses were conducted with IBM SPSS Statistics for Windows, Version 22.0 (IBM Corp., Armonk, NY, USA). Differences among the experimental groups were assessed using one-way analysis of variance (ANOVA), followed by Dunnett’s T3 post-hoc test. A *p*-value of less than 0.05 was considered indicative of statistical significance.

## 3. Results

### 3.1. Toxicity/Viability Assessment

To evaluate the effects of semaglutide on cell viability in NHDF and HREC cells, the MTT assay was performed. Cells subjected to oxidative stress and treated with semaglutide at concentrations of 11.25, 25.5, and 45 pg/mL for 24 h demonstrated that semaglutide not only mitigated H_2_O_2_-induced toxicity but also significantly increased cell viability compared to the control group. None of the tested concentrations exhibited cytotoxic effects; however, all concentrations significantly enhanced cell viability (*p* < 0.001) relative to the control (Figure 1A,B). Appendix A illustrates the assessment of cell viability in normal cells under conditions without oxidative stress.

### 3.2. Cell Death Assessment

The number of necrotic NHDF and HREC cells was subsequently assessed following exposure to semaglutide after oxidative stress. A cell counting assay was conducted to measure cell death. Treatment with 11.25, 25.5, and 45 pg/mL of semaglutide led to a decrease in cell death in NHDF and HREC cells compared to the control group (Figure 2A,B).

### 3.3. ATP Levels

To assess the impact of semaglutide on ATP levels in NHDF and HREC cells, ATP levels were measured using an ATP assay. Cells subjected to oxidative stress and treated with semaglutide at concentrations of 11.25, 25.5, and 45 pg/mL for 24 h showed that semaglutide not only alleviated H_2_O_2_-induced toxicity but also significantly promoted ATP production compared to the control group. All tested concentrations resulted in a significant increase in ATP levels (*p* < 0.001), indicating enhanced proliferation (Figure 3A,B). Appendix A depicts the measurement of ATP levels in normal cells under conditions without oxidative stress.

### 3.4. Intracellular ROS Production

To further investigate the mitochondrial apoptosis pathway activated by semaglutide under oxidative stress, ROS generation was assessed. The DCFH-DA fluorescent dye was used to quantify ROS levels in NHDF and HREC cells. As depicted in Figure 4A,B, treatment with semaglutide effectively reduced intracellular ROS production. Additionally, Figure 4C,D demonstrate that at all tested concentrations (11.25, 25.5, and 45 pg/mL), semaglutide significantly decreased mitochondrial superoxide production compared to the control group.

### 3.5. Mitochondrial Superoxide Production

To further elucidate the mitochondrial apoptosis pathway activated by semaglutide, mitochondrial superoxide generation was evaluated. As depicted in Figure 5A,B, treatment with semaglutide effectively reduced mitochondrial superoxide accumulation in NHDF and HREC cells. Additionally, Figure 5C,D shows that at all tested concentrations (11.25, 25.5, and 45 pg/mL), semaglutide significantly decreased mitochondrial superoxide production compared to the control group.

### 3.6. AGE and RAGE Expression

The impact of semaglutide on the production of AGEs and RAGE was subsequently investigated. Results demonstrated that treatment with semaglutide significantly reduced the levels of AGEs and RAGE in NHDF and HREC cells in a dose-dependent manner compared to the control group (Figure 6A–D).

### 3.7. Apoptosis Levels

The extent of apoptosis in NHDF and HREC cells was evaluated following treatment with semaglutide after oxidative stress induction. A cell counting assay demonstrated that treatment with semaglutide at concentrations of 11.25, 25.5, and 45 pg/mL significantly reduced apoptosis in both cell types compared to the control group (Figure 7A–D).

### 3.8. Antioxidant Response-Related Genes

The effect of semaglutide on the cellular antioxidant response was evaluated by measuring the gene expression levels of superoxide dismutase 1 (SOD1), catalase (CAT), glutathione peroxidase-1 (GPX1), and glutathione peroxidase-4 (GPX4) in NHDF and HREC cells. It was found that, under oxidative stress and after semaglutide treatment, these antioxidant genes were significantly upregulated compared to the stress control group (Figure 8A,B).

### 3.9. Semaglutide Increased Wound Closure Capacity in NHDFs

The observed increase in cell survival and proliferation in NHDF cells treated with semaglutide under oxidative stress prompted an investigation into its potential wound-healing effects on NHDFs. A wound-healing assay was performed on H_2_O_2_-treated NHDF cells, with wound closure monitored at 3, 6, 12, 24, and 36 h post-treatment. The percentage of wound area was measured to evaluate the wound-healing capacity of semaglutide. Results demonstrated that treatment with semaglutide significantly accelerated wound closure, with nearly complete closure observed at 24 h, as shown in Figure 9A,B.

### 3.10. Wound-Healing-Evaluation-Related Genes in NHDFs

The relative expression levels of genes involved in different stages of wound healing were evaluated to assess wound closure. The inflammatory phase was first examined by analyzing the expression of interleukin-1 beta (IL1B) and interleukin-2 (IL2). Additionally, genes associated with cell proliferation and extracellular matrix (ECM) remodeling were assessed to determine the effects of semaglutide under stress conditions. The expression of transcription factors such as epidermal growth factor (EGF) and fibroblast growth factor (FGF) was measured to evaluate cellular proliferation. For ECM-related processes, the expression levels of collagen types I, III, IV, and VI, specifically collagen type I alpha 1 chain (COL1A1), collagen type III alpha 1 chain (COL3A1), collagen type IV alpha 1 chain (COL4A1), and collagen type VI alpha 1 chain (COL6A1), as well as matrix metallopeptidases MMP3 and MMP9, were analyzed. Results indicated that, relative to the control group, the expression of IL1B and IL2 was significantly downregulated following semaglutide treatment (Figure 10). Conversely, upregulation was observed in the expression of EGF, FGF, and the collagen genes COL1A1, COL3A1, COL4A1, and COL6A1 (Figure 10). The expression levels of MMP3 and MMP9 were notably decreased under the same treatment conditions (Figure 10).

## 4. Discussion

This study demonstrates the significant protective and regenerative effects of semaglutide on NHDFs and HRECs subjected to oxidative stress, highlighting its potential as a therapeutic agent for conditions characterized by impaired wound healing, such as T2D. In the SUSTAIN 6 trial, a two-year cardiovascular outcomes study, semaglutide was linked to a significant increase in DR risk compared to the placebo [25]. Analyses suggest this may be mainly due to rapid and large reductions in HbA1c within the first 16 weeks, especially in patients with pre-existing DR, poor baseline control, and on insulin [32]. This early DR worsening is similar to what is seen with insulin therapy, linked to swift blood sugar improvements [32]. Current guidelines recommend close monitoring during insulin changes, and similar precautions should apply when using semaglutide to reduce DR progression risk [25,26,32].

Treatment with semaglutide markedly improves cellular viability and effectively reduces necrosis and apoptosis. As illustrated in Appendix A, semaglutide has a minimal impact on normal cells not subjected to oxidative stress. Additionally, cellular proliferation is stimulated, contributing to tissue regeneration. Semaglutide administration reduces intracellular ROS, leading to decreased levels of mitochondrial superoxide and overall oxidative stress within cells. This effect supports the preservation of mitochondrial membrane potential and function, which are essential for energy production and cellular survival. The suppression of ROS generation leads to diminished oxidative damage to cellular components, including lipids, proteins, and DNA, and consequently reduces cell death and dysfunction [33]. Taken together, the data demonstrate that semaglutide exhibits a strong antioxidant effect while showing no evidence of carcinogenic potential.

Concurrently, upregulation of the expression of endogenous antioxidant defense genes, namely SOD1, CAT, GPX1, and GPX4, is observed following semaglutide administration. This enhancement of the antioxidant response indicates that the capacity for ROS neutralization is boosted, which further mitigates oxidative stress. The increased activity of these enzymes is critical for maintaining redox homeostasis and protecting cells from oxidative insults.

In addition to the antioxidative effects, the clearest evidence of modulation within glycation pathways is provided by the significant decrease in levels of AGEs and RAGE expression. These modifications are known to promote oxidative stress, inflammation, and tissue damage, and their reduction strongly suggests that semaglutide interrupts the vicious cycle of glycation and oxidative injury, which is particularly pertinent in diabetic tissues [34]. Consequently, the accumulation of AGEs is minimized, tissue elasticity and function are preserved, and the progression of diabetic microvascular and macrovascular complications is potentially mitigated [35].

A previous study using the db/db mouse model evaluated the impact of topical semaglutide on retinal inflammation and microvascular damage, in response to concerns regarding its direct effects on the retina [36]. Over a period of 15 days, eye drops containing semaglutide or vehicle were administered. It was found that glial activation, the expression of inflammatory cytokines, NF-κB levels, and ICAM-1 were significantly reduced by semaglutide. Retinal ganglion cell apoptosis was prevented, and activation of the AKT pathway was observed. A substantial decrease in vascular leakage was also noted. These effects were seen without any change in blood glucose levels, indicating that they can be attributed directly to semaglutide [36].

The anti-inflammatory potential of semaglutide is supported by the observed downregulation of key cytokines such as IL1B and IL2. Chronic inflammation is recognized as a detrimental factor that prolongs the inflammatory phase of wound healing, delays tissue regeneration, and leads to tissue breakdown [37]. The suppression of these cytokines indicates that semaglutide may effectively reduce inflammatory signaling, thereby facilitating progression to subsequent healing phases [38].

Gene expression analyses further reveal that the treatment results in the upregulation of critical growth factors involved in tissue repair, such as EGF and FGF. These factors stimulate fibroblast proliferation, collagen synthesis, and re-epithelialization. Significantly, the expression levels of various ECM components including collagen types I, III, IV, and VI (COL1A1, COL3A1, COL4A1, COL6A1) are increased, indicating enhanced matrix deposition and stabilization, which are crucial for wound strength and integrity. Conversely, the expression levels of matrix metalloproteinases MMP3 and MMP9 are decreased, reducing excessive ECM degradation and supporting proper tissue remodeling.

The healing process of diabetic wounds is hindered by the decreased proliferation and migration of epidermal and endothelial cells [39]. A semaglutide-loaded nanofiber membrane was developed to promote diabetic wound healing by suppressing ROS production and accelerating angiogenesis, using a diabetic rat model. The membrane was prepared through an electrostatic spinning technique, consisting of a thermoplastic polyurethane elastomer, a polyvinyl butyral ester, zein, and semaglutide. It was characterized by a uniform diameter, appropriate water vapor transmission rate, and effective water absorption and retention properties. In vivo, it was observed that the nanofiber membrane loaded with semaglutide facilitated the formation of a normal skin tissue structure during regeneration. In vitro, it was shown that semaglutide might help mitigate high-glucose-induced endothelial cell dysfunction by inhibiting ROS production. The study thus provided a theoretical foundation for the potential application of semaglutide in the treatment of diabetic ulcers and other wounds [39].

The functional relevance of these molecular changes is demonstrated by the wound-healing assay, where the rate of wound closure is significantly accelerated by semaglutide treatment. The rapid migration and proliferation of fibroblasts, along with ECM deposition, contribute to the prompt closure of wounds, emphasizing semaglutide’s potential benefits for improving healing outcomes, especially in chronic wound environments typical of patients with T2D [40].

Furthermore, GLP-1 receptor activation enhances cellular antioxidative defenses by increasing the expression of enzymes such as SOD1, CAT, GPX1, and GPX4, thereby reducing the accumulation of ROS. This activation also leads to a reduction in the production of pro-inflammatory cytokines, including IL-1β and IL-6. Additionally, the activation of FGF and EGF signaling pathways promotes ECM synthesis, supporting tissue repair and remodeling.

Importantly, these mechanisms are further supported by the modulation of pathways involved in the formation and accumulation of AGEs, which are known to exacerbate oxidative stress and inflammation in chronic conditions. By reducing AGE formation and oxidative stresses, semaglutide may attenuate the deleterious cellular effects associated with glycation and oxidative damage, thereby promoting regenerative processes and cellular resilience.

This study represents, to the best of our knowledge, the first investigation into the effects of semaglutide on human fibroblasts and retinal endothelial cells, emphasizing its potential novel role in promoting cellular protection and tissue regeneration.

Future research should focus on optimizing dosing to enhance efficacy and reduce side effects during long-term use. Long-term safety studies are essential, especially for chronic conditions like T2D. In vivo studies using animal models that mimic human diabetic wounds are vital to understand semaglutide’s pharmacodynamics and pharmacokinetics within complex systems. Exploring combination therapies, such as with bioengineered scaffolds or growth factors, may improve healing outcomes, particularly in difficult wounds. Additional research should also investigate semaglutide’s potential in other chronic diseases involving oxidative stress, like cardiovascular conditions and neurodegeneration. Mechanistic studies are needed to clarify the signaling pathways involved, which could guide targeted therapies and biomarker development. For clinical translation, early trials should assess safety, dosing, and initial efficacy in patients with diabetic or chronic wounds to determine its practical use and suitable patient populations.

## 5. Conclusions

Semaglutide demonstrates potent antioxidant effects in human dermal fibroblasts and retinal endothelial cells by reducing oxidative stress, improving cell viability, and decreasing apoptosis. Its ability to upregulate endogenous antioxidant genes and downregulate pro-inflammatory mediators further supports its protective role against oxidative damage. Additionally, semaglutide accelerates wound closure by promoting cell proliferation, migration, and extracellular matrix remodeling, which are critical processes in tissue regeneration. These combined properties of semaglutide highlight its potential as a promising therapeutic agent for the management of diabetic foot ulcers, where impaired healing is often driven by oxidative stress and inflammation, as well as for addressing other oxidative stress-related complications in people with diabetes like DR. Further in vivo research is needed to investigate the potential effect of semaglutide for the prevention and treatment of diabetes-related complications.

## Figures and Tables

**Figure 1 pharmaceutics-17-01115-f001:**
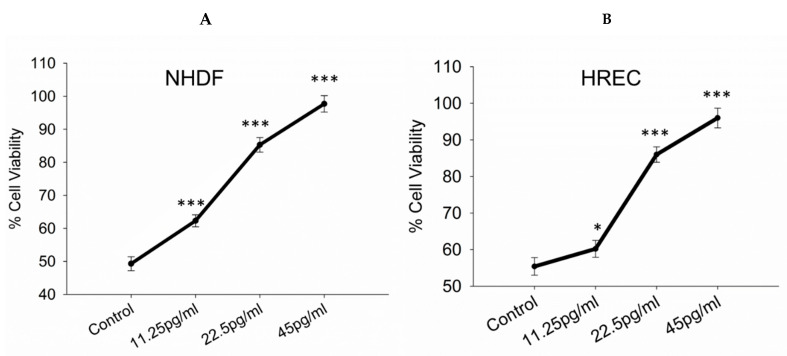
Assessment of cell viability. (**A**,**B**) Cell viability levels of NHDFs and HRECs were assessed using the MTT assay. Cells were cultured with varying concentrations of semaglutide (control: medium only; 11.25, 25.5, and 45 pg/mL) for 24 h. The results are expressed as the mean ± SD from three independent experiments (*n* = 3). Statistical analysis was performed using one-way ANOVA followed by Dunnett’s T3 post-hoc test. Significance levels are indicated as * *p* < 0.05 and *** *p* < 0.001 when compared to the control group. ANOVA, analysis of variance; HRECs, human retinal endothelial cells; MTT, 3-(4,5-dimethylthiazol-2-yl)-2,5-diphenyltetrazolium bromide; NHDFs, normal human dermal fibroblasts; SD, standard deviation.

**Figure 2 pharmaceutics-17-01115-f002:**
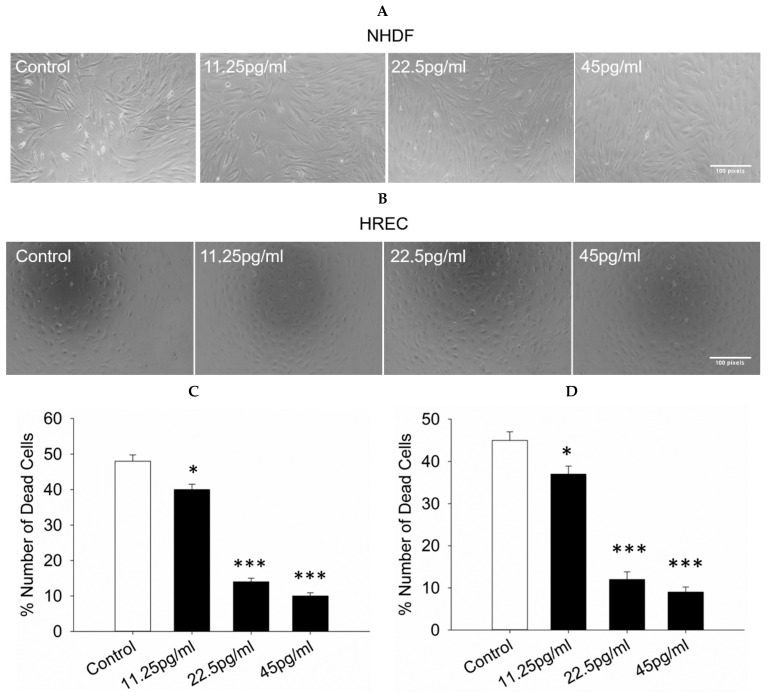
Quantification of dead cells. The number of dead cells among NHDFs and HRECs was assessed after culturing with semaglutide at various concentrations (control: medium only; 11.25, 25.5, and 45 pg/mL) for 24 h. (**A**,**B**) Images were acquired at 5× magnification, with a scale bar representing 100 pixels. Cell death was assessed using the trypan blue exclusion assay. (**C**,**D**) Data are presented as the mean ± SD from three independent experiments (*n* = 3). Statistical analysis was conducted using one-way ANOVA followed by Dunnett’s T3 post-hoc test. Statistical significance is indicated by * *p* < 0.05 and *** *p* < 0.001 compared to the control group. ANOVA, analysis of variance; HRECs, human retinal endothelial cells; NHDFs, normal human dermal fibroblasts; SD, standard deviation.

**Figure 3 pharmaceutics-17-01115-f003:**
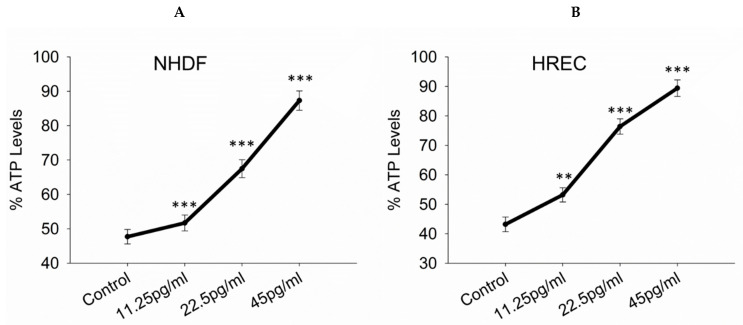
Measurement of ATP levels in NHDFs and HRECs. (**A**,**B**) ATP levels as determined using the Vialight Plus assay kit. NHDFs and HRECs were cultured with semaglutide at various concentrations (control: medium only; 11.25, 25.5, and 45 pg/mL) for 24 h. The data are expressed as the mean ± SD of three independent experiments (*n* = 3). Statistical significance was determined via one-way ANOVA followed by the Dunnett’s T3 post-hoc test, where ** *p*-value < 0.01, and *** *p*-value < 0.001, in comparison with the control group. ANOVA, analysis of variance; ATP, adenosine triphosphate; HRECs, human retinal endothelial cells; NHDFs, normal human dermal fibroblasts; SD, standard deviation.

**Figure 4 pharmaceutics-17-01115-f004:**
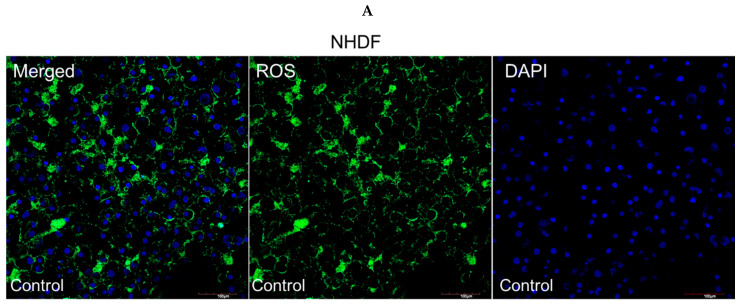
Evaluation of intracellular ROS levels. (**A**,**B**) NHDF and HREC cultures were treated with varying concentrations of semaglutide (control: medium only; 11.25, 25.5, and 45 pg/mL) for 24 h. The left column shows overlay images of ROS with DAPI staining, the middle column displays ROS-specific fluorescence (green), and the right column shows nuclear DAPI staining (blue). Images were captured at 20× magnification, with a scale bar of 100 μm. (**C**,**D**) The bar graphs display the quantitative analysis of ROS levels, presented as the mean ± SD from three independent experiments (*n* = 3). Statistical significance was assessed using one-way ANOVA followed by Dunnett’s T3 post-hoc test. Significance levels are indicated as ** *p* < 0.01, and *** *p* < 0.001, compared to the control group. ANOVA, analysis of variance; DAPI, 4′,6-diamidino-2-phenylindole; DCFDA, 2′,7′-dichlorofuorescein diacetate; HRECs, human retinal endothelial cells; NHDFs, normal human dermal fibroblasts; ROS, reactive oxygen species; SD, standard deviation.

**Figure 5 pharmaceutics-17-01115-f005:**
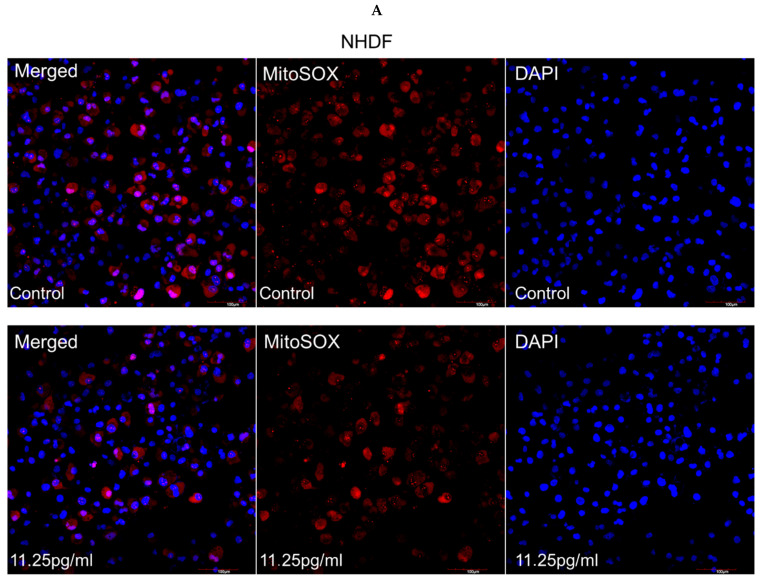
Assessment of mitochondrial superoxide production. (**A**,**B**) Mitochondrial superoxide production was evaluated using MitoSOΧ™. NHDF and HREC cultures were treated with varying concentrations of semaglutide (control: medium only; 11.25, 25.5, and 45 pg/mL) for 24 h. The left column shows overlay images of MitoSOΧ™ with DAPI staining, the middle column displays MitoSOΧ™-specific fluorescence (red) indicating mitochondrial superoxide, and the right column shows nuclear DAPI staining (blue). All images were captured at 20× magnification, with a scale bar of 100 μm. (**C**,**D**) The bar graphs depict quantitative data, expressed as the mean ± SD from three independent experiments (*n* = 3). Statistical significance was determined using one-way ANOVA followed by Dunnett’s T3 post-hoc test. Significance levels are indicated as * *p* < 0.05, ** *p* < 0.01, and *** *p* < 0.001 compared to the control group. ANOVA, analysis of variance; DAPI, 4′,6-diamidino-2-phenylindole; HRECs, human retinal endothelial cells; MitoSOΧ, mitochondrial superoxide indicator; NHDFs, normal human dermal fibroblasts; SD, standard deviation.

**Figure 6 pharmaceutics-17-01115-f006:**
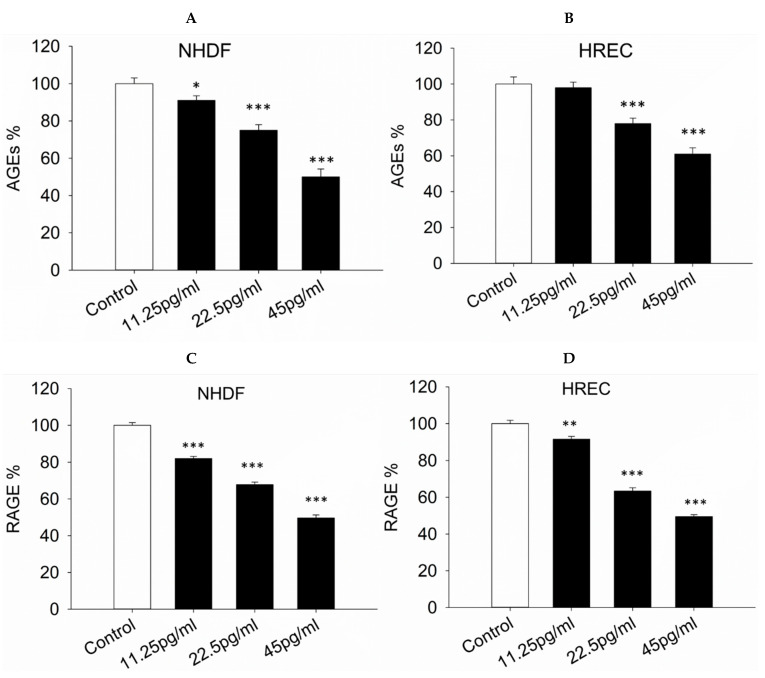
Concentration of AGEs and RAGE in cell culture supernatants. (**A**–**D**) Intracellular levels of AGEs and RAGE concentrations were measured using a colorimetric assay kit. NHDF and HREC cells were cultured with varying concentrations of semaglutide (control: medium only; 11.25, 25.5, and 45 pg/mL) for 24 h. The data are expressed as the mean ± SD from three independent experiments (*n* = 3). Statistical significance was assessed using one-way ANOVA followed by Dunnett’s T3 post hoc test. Significance levels are indicated as * *p* < 0.05, ** *p* < 0.01, and *** *p* < 0.001 when compared to the control group. AGEs, advanced glycation end-products; ANOVA, analysis of variance; HRECs, human retinal endothelial cells; NHDFs, normal human dermal fibroblasts; RAGE, receptor for AGEs; SD, standard deviation.

**Figure 7 pharmaceutics-17-01115-f007:**
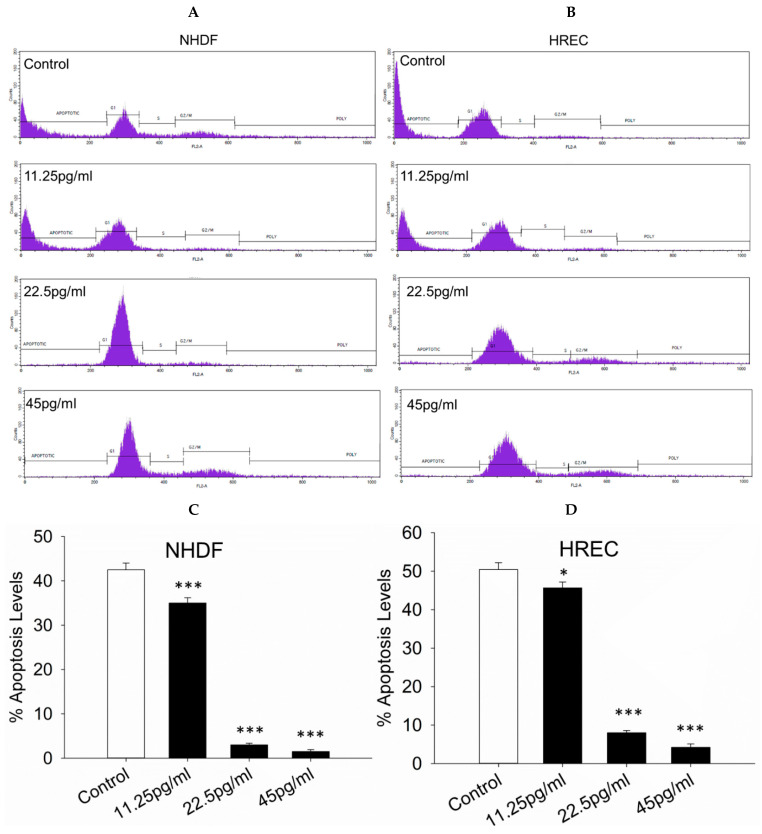
Assessment of apoptosis levels. (**A**,**B**) The histograms illustrate the levels of apoptosis and cell cycle phases. These parameters were assessed via flow cytometry using propidium iodide (PI) staining. (**C**,**D**) The bar graph depicts the quantitative analysis of apoptosis in NHDF and HREC cells cultured with various concentrations of semaglutide (control: medium only; 11.25, 25.5, and 45 pg/mL) for 24 h. Data are expressed as the mean ± SD of three independent experiments (*n* = 3). Statistical significance was evaluated using one-way ANOVA followed by Dunnett’s T3 post-hoc test. Significance levels are indicated as * *p* < 0.05 and *** *p* < 0.001, compared to the control group. ANOVA, analysis of variance; HRECs, human retinal endothelial cells; NHDFs, normal human dermal fibroblasts; PI, propidium iodide; SD, standard deviation.

**Figure 8 pharmaceutics-17-01115-f008:**
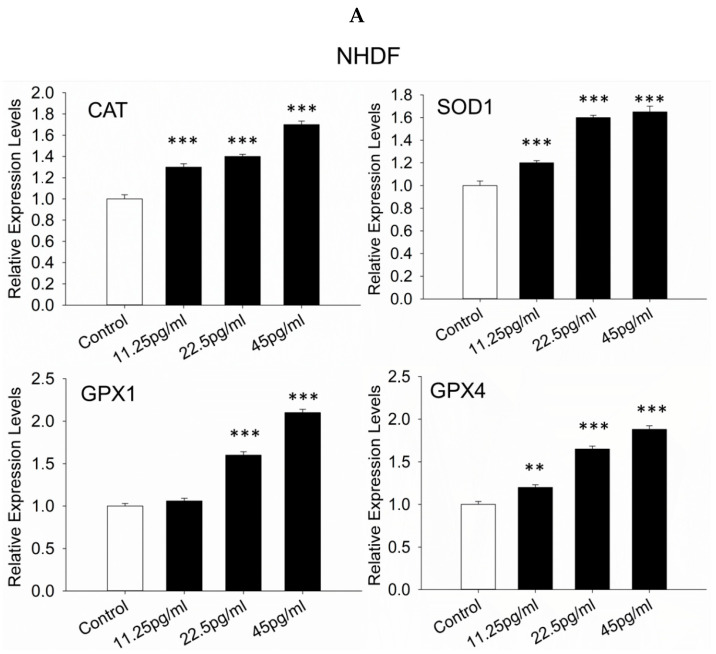
Relative expression levels of SOD1, CAT, GPX1, and GPX4 in NHDFs and HRECs. (**A**,**B**) Cells were cultured with varying concentrations of semaglutide (control: medium only; 11.25, 25.5, and 45 pg/mL) for 24 h. ACTB was used as the internal reference gene. The data are presented as the mean ± SD from three independent experiments (*n* = 3). Statistical significance was assessed using one-way ANOVA followed by Dunnett’s T3 post-hoc test. Significance levels are indicated as ** *p* < 0.01, and *** *p* < 0.001 compared to the control group. ACTB, actin beta; ANOVA, analysis of variance CAT, catalase; GPX1, glutathione peroxidase; GPX4, glutathione peroxidase-4; NHDFs, normal human dermal fibroblasts; SD, standard deviation; SOD1, superoxide dismutase 1.

**Figure 9 pharmaceutics-17-01115-f009:**
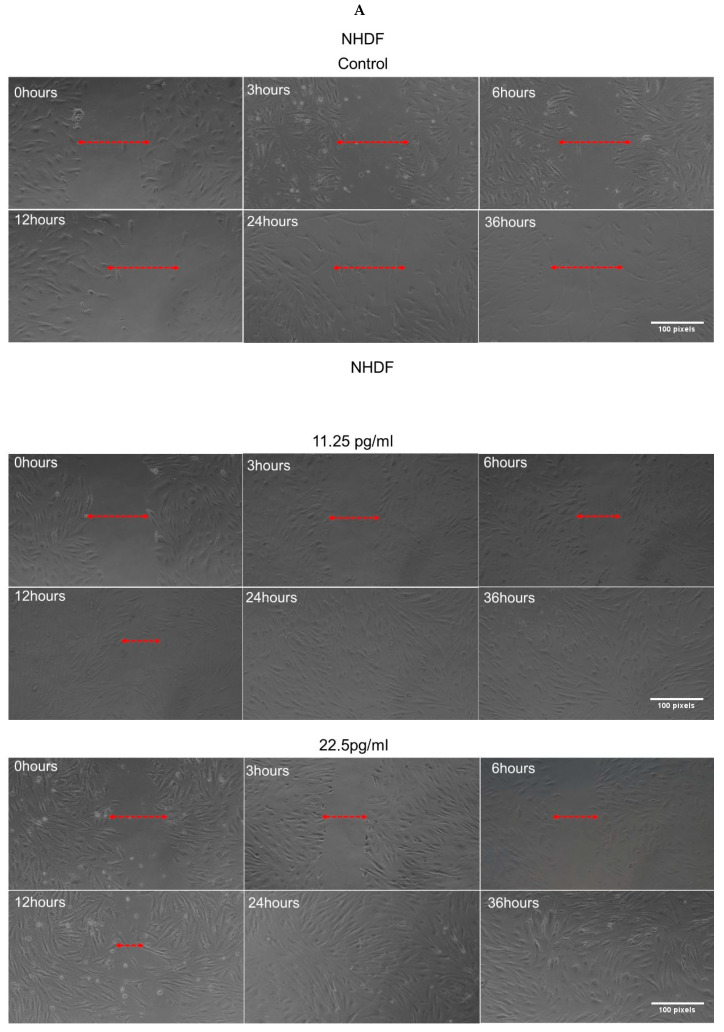
Percent NHDF wound area. A wound scratch assay was performed on NHDF cells cultured with semaglutide at various concentrations (control: medium only; 11.25, 25.5, and 45 pg/mL) for 24 h. (**A**) The assay images show the progression of wound closure at different time points (3, 6, 12, 24, and 36 h). Horizontal arrows indicate the width of the wound at each time point, demonstrating significant wound closure in NHDF cells treated with semaglutide. Quantitative analysis of wound closure was performed using ImageJ software, with images captured at 5× magnification; scale bar = 100 pixels. (**B**) The bar graph presents the percentage of wound closure across different treatment groups. Results show that semaglutide significantly accelerates wound healing compared to the control group. Data are expressed as the mean ± SD from three independent experiments (*n* = 3). Statistical significance was determined via one-way ANOVA followed by Dunnett’s T3 post-hoc test, with significance levels indicated as *** *p* < 0.001 compared to the control. ANOVA, analysis of variance; ATP, adenosine triphosphate; NHDFs, normal human dermal fibroblasts; SD, standard deviation.

**Figure 10 pharmaceutics-17-01115-f010:**
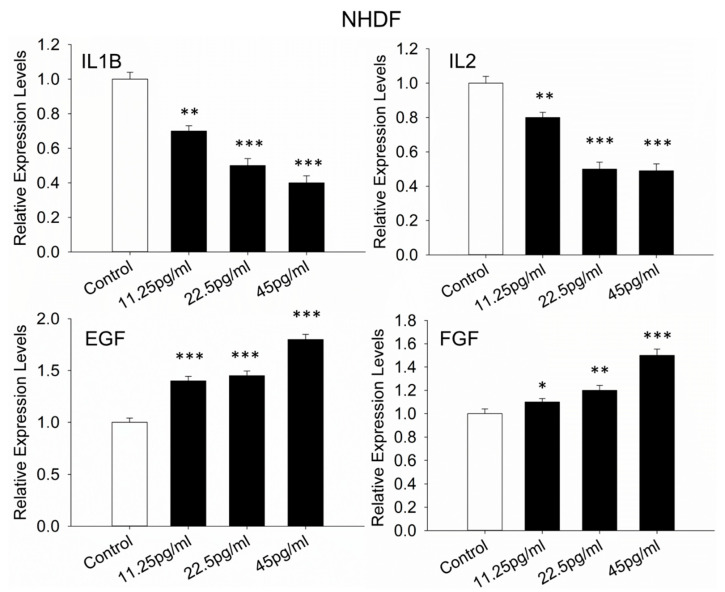
Relative expression levels. Relative expression levels of IL1B, IL2, EGF, FGF, COL1A1, COL3A1, COL4A1, COL6A1, MMP3, and MMP9 were assessed in NHDFs following 24 h treatment with varying concentrations of semaglutide. The treatment groups included a control (medium only) and semaglutide at concentrations of 11.25, 25.5, and 45 pg/mL. ACTB was used as the internal reference gene. Data are expressed as the mean ± SD from three independent experiments (*n* = 3). Statistical analysis was performed using one-way ANOVA followed by Dunnett’s T3 post-hoc test. Significance levels were denoted as * *p* < 0.05, ** *p* < 0.01, and *** *p* < 0.001, compared to the control group. ACTB, actin beta; ANOVA, analysis of variance; COL1A1, collagen type I alpha 1 chain; COL3A1, collagen type III alpha 1 chain; COL4A1, collagen type IV alpha 1 chain; COL6A1, collagen type VI alpha 1 chain; EGF, epidermal growth factor; FGF, fibroblast growth factor; IL1B, interleukin 1 beta; IL2, interleukin 2; MMP3, matrix metallopeptidase 3; MMP9, matrix metallopeptidase 9; NHDFs, normal human dermal fibroblasts; SD, standard deviation.

## Data Availability

The raw data supporting the conclusions of this article will be made available by the authors on request.

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
