# Peer review of "Semaglutide Enhances Cellular Regeneration in Skin and Retinal Cells In Vitro"

_pharmaceutics, 2025, doi:10.3390/pharmaceutics17091115_

Round 1

Reviewer 1 Report

Comments and Suggestions for Authors

The present study examines the effects of Semaglutide on the regeneration of human dermal fibroblasts and retinal endothelial cells. Semaglutide has long been utilized in the treatment of certain types of diabetes, and extensive research has been conducted on its therapeutic applications. By evaluating various variables, the study has demonstrated that this drug exhibits promising regenerative properties when interacting with skin and retinal cells. However, despite the valuable findings, several limitations exist in the study’s design and methodology. It is advisable for the authors to consider the following points:

  1. The specific objective of the study is not clearly articulated. Is the focus on the drug’s efficacy in diabetic patients, or is its potential being explored independently of diabetes? Given the drug's mechanism of action and its established use in diabetes treatment, the processes of cellular damage and repair differ significantly in diabetic individuals and are not solely linked to oxidative stress. Consequently, investigating the effects of this drug on normal cells may not accurately reflect its regenerative capacity in diabetics. This distinction should be clarified.
  2. The criteria for selecting the drug concentrations to assess its efficacy are not adequately explained. Further justification for the chosen concentrations would enhance the study’s rigor.
  3. Figure 1 indicates an approximate 200% increase in cell viability under oxidative stress within 24 hours. While this finding suggests enhanced cellular repair and regeneration, it may also imply a potential carcinogenic effect. A thorough review of these results is recommended. Similarly, Figure 3 demonstrates a significant increase in ATP content, which warrants further scrutiny.
  4. The discussion section lacks depth and does not sufficiently contextualize or justify the results obtained. A more comprehensive analysis is necessary to support the study's conclusions.

Author Response

13 August 2025

Dear Editor,

please find enclosed our REVISED manuscript entitled “Semaglutide Enhances Cellular Regeneration in Skin and Retinal Cells In Vitro” to be considered for publication. We would like to thank you and the reviewers for your thoughtful evaluation of our manuscript and for your most welcome comments/suggestions, through which we believe that we have improved the quality of our manuscript. Accordingly, we have now revised our manuscript thoroughly to reflect these comments.

Please find below a point-by-point response to ALL the issues raised by the Reviewers. All changes in the main text are marked in red color.

Reviewing: 1

Comment 1: The specific objective of the study is not clearly articulated. Is the focus on the drug’s efficacy in diabetic patients, or is its potential being explored independently of diabetes? Given the drug's mechanism of action and its established use in diabetes treatment, the processes of cellular damage and repair differ significantly in diabetic individuals and are not solely linked to oxidative stress. Consequently, investigating the effects of this drug on normal cells may not accurately reflect its regenerative capacity in diabetics. This distinction should be clarified.

Our Response. First, we would like to thank the reviewer for this comment. The manuscript has been revised as follows: The primary objective of this study was to examine whether semaglutide has cytoprotective effects against oxidative stress in human dermal fibroblasts (NHDF) and human retinal endothelial cells (HREC). Specifically, the study aimed to assess the effect of semaglutide on key aspects of wound healing in fibroblasts, including cell migration and the expression profiles of extracellular matrix-associated genes. Additionally, the research sought to determine whether semaglutide enhances cellular viability, attenuates apoptosis, and mitigates oxidative damage in response to hydrogen peroxide–induced oxidative stress. Furthermore, the study examined whether semaglutide exerts direct protective effects on retinal endothelial cells, thereby suggesting a broader potential therapeutic role in DR that extends beyond its established glycemic regulation. These findings aim to contribute to a deeper understanding of semaglutide’s cellular mechanisms and its prospective application in vascular and tissue repair within diabetic retinal pathology. (Section: Introduction last paragraph).

Comment 2: The criteria for selecting the drug concentrations to assess its efficacy are not adequately explained. Further justification for the chosen concentrations would enhance the study’s rigor.

Our Response: Thank you for this comment. We have addressed the issue accordingly and included the appropriate references. ‘Semaglutide concentrations used in this study were based on those reported in a previous study [28,29].’ (Section: Methods last paragraph).

Comment 3: Figure 1 indicates an approximate 200% increase in cell viability under oxidative stress within 24 hours. While this finding suggests enhanced cellular repair and regeneration, it may also imply a potential carcinogenic effect. A thorough review of these results is recommended. Similarly, Figure 3 demonstrates a significant increase in ATP content, which warrants further scrutiny.

Our Response. Thank you for this comment. The control group demonstrates 100% viability and serves as the baseline for comparing oxidative stress conditions. Compared to unstressed cells, which showed only a 5-7% increase in viability and ATP levels, our results do not indicate any evidence of carcinogenic effects. These data are included as supplementary material (Figure 1,2) for further clarification.

Comment 4: The discussion section lacks depth and does not sufficiently contextualize or justify the results obtained. A more comprehensive analysis is necessary to support the study's conclusions.

Our Response: The discussion section has been extensively revised according to your suggestion, and it is now focused on the results of our experiments. In addition, we compare the results of our study with others in literature.

Kind regards,

Reviewer 2 Report

Comments and Suggestions for Authors

The manuscript by IA Anastasiou et al., entitled "Beyond Glycemic Control: Semaglutide Supports Cellular Regeneration in Skin and Retina”, reports the antioxidative and cytoprotective effects of the glucagon-like peptide-1 receptor agonist (semaglutide) on cells under hydrogen peroxide (Hâ‚‚Oâ‚‚)-induced oxidative stress. Human dermal fibroblasts and retinal endothelial cells were treated with different concentration of semaglutide and cell viability, intracellular reactive oxygen species (ROS) levels, and cell migration were assessed. Although the Authors presented interesting results to the field, some minor shortfalls should be addressed.

Comments for Authors:

The title of the article "Beyond Glycemic Control: Semaglutide Supports Cellular Regeneration in Skin and Retina” is misleading. The study was performed on cell lines in vitro, not in tissues.

The Material and Methods section:

- “Detection of cell apoptosis” subhead: Please designate the method used for detection of cells stained with propidium iodide.

The Result section:

  • “Proliferation/ATP levels” subhead, p. 7: Authors wrote that “semaglutide not only alleviated Hâ‚‚Oâ‚‚-induced toxicity but also significantly promoted cell proliferation compared to the control group”. This statement is unsubstantiated since the study did not use any tests that assessed cell proliferation.
  • Figure 7, A, B: “The dot plot illustrates the levels of apoptosis and cell cycle phases”: In fact, the graphs are presented as histograms.

Author Response

13 August 2025

Dear Editor,

please find enclosed our REVISED manuscript entitled “Semaglutide Enhances Cellular Regeneration in Skin and Retinal Cells In Vitro” to be considered for publication. We would like to thank you and the reviewers for your thoughtful evaluation of our manuscript and for your most welcome comments/suggestions, through which we believe that we have improved the quality of our manuscript. Accordingly, we have now revised our manuscript thoroughly to reflect these comments.

Please find below a point-by-point response to ALL the issues raised by the Reviewers. All changes in the main text are marked in red color.

Reviewing: 2

Comments and Suggestions for Authors

The manuscript by IA Anastasiou et al., entitled "Beyond Glycemic Control: Semaglutide Supports Cellular Regeneration in Skin and Retina”, reports the antioxidative and cytoprotective effects of the glucagon-like peptide-1 receptor agonist (semaglutide) on cells under hydrogen peroxide (Hâ‚‚Oâ‚‚)-induced oxidative stress. Human dermal fibroblasts and retinal endothelial cells were treated with different concentration of semaglutide and cell viability, intracellular reactive oxygen species (ROS) levels, and cell migration were assessed. Although the Authors presented interesting results to the field, some minor shortfalls should be addressed.

Comments for Authors:

Comment 1: The title of the article "Beyond Glycemic Control: Semaglutide Supports Cellular Regeneration in Skin and Retina” is misleading. The study was performed on cell lines in vitro, not in tissues.

Our Response. We agree with your comment. We have made the necessary changes accordingly. ‘Semaglutide Enhances Cellular Regeneration in Skin and Retinal Cells In Vitro’.

Comment 2: - “Detection of cell apoptosis” subhead: Please designate the method used for detection of cells stained with propidium iodide.

Our Response. Thank you for this comment. We have made the following changes in these sections as follows: ‘Propidium iodide staining is used for detecting apoptosis NHDF and HREC cells were seeded in 6-well plates in triplicate and then treated with semaglutide for 24 hours. Following treatment, cells were harvested and resuspended in 500 μl of PBS. To induce cell permeabilization, 4.5 ml of 80% ethanol was added to the cell suspension, and samples were incubated at 4°C for 20 minutes. Cells were then centrifuged at 200 x g for five minutes at 4°C, and the supernatant was carefully removed. The cell pellet was washed twice with cold PBS, with each wash followed by centrifugation at 200 x g for five minutes to remove residual ethanol. Next, cells were permeabilized by incubation with 0.1% Triton X-100 for five minutes, followed by centrifugation to remove the supernatant. The cells were resuspended in 100 μl of propidium iodide (PI) staining solution, consisting of PI (4 μg/ml), ribonuclease (0.1 mg/ml), Tris-HCl buffer (pH 7.5, 100 mM), NaCl (150 mM), CaClâ‚‚ (1 mM), MgClâ‚‚ (0.5 mM), and NP-40 (0.1%). The stained cells were incubated at room temperature for 30 minutes in the dark to prevent photobleaching. Following incubation, 400 μl of PBS was added to each sample, and flow cytometric analysis was performed using a BD FACSCalibur (BD Biosciences, San Jose, CA, USA). Data were acquired from the FL2 channel, and the percentage of PI-positive (i.e., apoptotic or dead) cells was calculated as a proportion of the total cell population: (PI-positive cells / total cells) × 100%.’ (Section: Methods)

Comment 3: “Proliferation/ATP levels” subhead, p. 7: Authors wrote that “semaglutide not only alleviated Hâ‚‚Oâ‚‚-induced toxicity but also significantly promoted cell proliferation compared to the control group”. This statement is unsubstantiated since the study did not use any tests that assessed cell proliferation.

Our Response. We have made the necessary improvements. (Section: Results). ‘ATP levels. To assess the impact of semaglutide on ATP levels in NHDF and HREC cells, ATP levels were measured using an ATP assay. Cells subjected to oxidative stress and treated with semaglutide at concentrations of 11,25, 25,5, and 45 pg/ml for 24 hours showed that semaglutide not only alleviated Hâ‚‚Oâ‚‚-induced toxicity but also significantly promoted ATP production compared to the control group. All tested concentrations resulted in a significant increase in ATP levels (p < 0.001), indicating enhanced proliferation (Fig. 3A, B). Supplementary Figure 2 depicts the measurement of ATP levels in normal cells under conditions without oxidative stress.

Comment 4: Figure 7, A, B: “The dot plot illustrates the levels of apoptosis and cell cycle phases”: In fact, the graphs are presented as histograms.

Our Response. We have addressed the issue accordingly. (Section: Results ). ‘The histograms illustrate the levels of apoptosis and cell cycle phases. These parameters were assessed via flow cytometry using propidium iodide (PI) staining’.

Kind regards,

Round 2

Reviewer 1 Report

Comments and Suggestions for Authors

The authors' responses to most of the comments were convincing. However, they need to revise how they report cell viability and the percentage increase in ATP. It appears they used the viability of non-oxidative stress cells as the baseline, which makes the figures unclear and the graphs exaggerated. Please adjust the y-axis in both Figures 1 and 3 to the standard range of 1 to 100 percent for reporting percentages.

Author Response

17 August 2025

Dear Editor,

please find enclosed our REVISED manuscript entitled “Semaglutide Enhances Cellular Regeneration in Skin and Retinal Cells In Vitro” to be considered for publication. We would like to thank you and the reviewers for your thoughtful evaluation of our manuscript and for your most welcome comments/suggestions, through which we believe that we have improved the quality of our manuscript. Accordingly, we have now revised our manuscript thoroughly to reflect these comments.

Please find below a point-by-point response to ALL the issues raised by the Reviewers. All changes in the main text are marked in red color.

Reviewing: 1

Reviewer 1

Comment 1:The authors' responses to most of the comments were convincing. However, they need to revise how they report cell viability and the percentage increase in ATP. It appears they used the viability of non-oxidative stress cells as the baseline, which makes the figures unclear and the graphs exaggerated. Please adjust the y-axis in both Figures 1 and 3 to the standard range of 1 to 100 percent for reporting percentages.

Response 1: We appreciate the reviewer's insightful comment. Figures 1 and 3 have been updated to include the standard range for better clarity and comparison.